

# Movements of post-breeding royal terns (*Thalasseus maximus*) in Virginia, U.S.A

Chelsea Weithman[1], Daniel Catlin[1], Sarah Karpanty[1], Kelsi Hunt[1], Camille Alvino[1], Joanna Morelli[1], Will Britton[1], Ruth Boettcher[2], Rebecca Gwynn[2], Michael Guilfoyle[3] and James Fraser[1]

[1] Department of Fish and Wildlife Conservation, Virginia Polytechnic Institute and State University (Virginia Tech), Blacksburg, VA, United States of America
[2] Virginia Department of Wildlife Resources, Henrico, VA, United States of America
[3] United States Army Corps of Engineers, Vicksburg, MS, United States of America

## ABSTRACT

Seabirds use a variety of strategies to maximize survival during migration. We studied the post-incubation movements of royal terns (*Thalasseus maximus*) from the largest seabird colony in Virginia, U.S.A. to understand their behaviors after nesting. We affixed GPS/GSM transmitters to nine incubating adult royal terns near the end of incubation (late June 2022) until migration (early December 2022). We used a hidden Markov model to describe royal tern behavior from late incubation through part of the migration season. Royal terns were classified as either resting or exploratory (*e.g.*, flying, foraging) based on their speed and turning angles. Royal terns spent most of their time in the exploratory state (0.711, 95% CI [0.710–0.711]). Royal terns tended to move north following the breeding season before migrating to the south in October and November. We speculate that the post-breeding movement north was to track Atlantic menhaden (*Brevoortia tyrranus*), a primary food source. As threats to royal terns on the Atlantic coast such as declining prey fish stocks and offshore energy development increase, knowledge of their behaviors and movements will be essential to their conservation.

Corresponding author
Daniel Catlin, dcatlin@vt.edu

## INTRODUCTION

Habitat use and movements between the end of breeding and departure for non-breeding areas are unknown for many avian species. Some bird species use staging locations to prepare for long migrations or stopovers for relatively shorter trips, with the length of stay and quality of the habitat being related to the type of migration (*Warnock, 2010*). The lack of a central place like a nest or a consistent roost location makes it difficult for researchers to study this period in the annual cycle.

The royal tern (*Thalasseus maximus*) is one of the largest terns in the world (*Buckley, Buckley & Mlodinow, 2021*). Royal terns nest in densely-packed colonies, typically lay a single egg, and the young form crèches soon after hatching (*Buckley, Buckley & Mlodinow, 2021*). Little is known about the post-breeding movements of royal terns. Banding records suggest that individuals from the mid-Atlantic coast of the United States tend to move north to stage before migrating southward to coastal wintering grounds as far south as Peru

(*Buckley, Buckley & Mlodinow, 2021*; *Gibson et al., 2023*). In general, migration strategies in birds tend to minimize the amount of time spent migrating (*Alerstam & Lindström, 1990*), which appears to contradict with the northward movement of post-breeding royal terns. However, the pre-migratory period can be critical to survival during migration, and the strategies (*e.g.*, time minimization, frequent stops, *etc.*) used to deal with this period vary among seabirds (*Loring et al., 2017*). It is possible that royal terns may capitalize on a resource or modify other behaviors in a way that would mitigate the overall cost of a delayed or counter-migration (*Klaassen et al., 2012*). Capitalizing on abundant food resources that occur opposite of the main direction of travel may allow individuals to adequately fuel for the journey (*Lindström et al., 2011*). It may also provide access to wind currents and more favorable migratory routes to the destination, or avoid migration routes of potential avian predators (*Newton, 2010*), such as peregrine falcons (*Falco peregrinus*).

Royal tern numbers declined throughout the mid-Atlantic in the 1990s and early 2000s (*Brinker et al., 2007*), as did survival rates of all age classes, although all age classes appear to be rebounding (*Gibson et al., 2023*). Commercial fishing, wind power development and climate change are of great concern for many seabird species (*Croxall et al., 2012*; *Searle et al., 2023*), including royal terns. Evidence from long-term banding of royal terns suggests that increasing temperatures, sea-levels, and continued fishing pressure may threaten the species (*Gibson et al., 2023*). Specifically, royal tern survival was negatively correlated with landings of Atlantic menhaden (*Brevoortia tyrannus*), a commercially and ecologically important fish species that also migrates along the Atlantic Coast and in the Chesapeake Bay (*Anstead et al., 2021*; *Gibson et al., 2023*; *Liljestrand, Wilberg & Schueller, 2019*). Additionally, survival of pre-breeding royal terns (age 0–2) has decreased 55% since the 1960's, co-influenced by rising sea surface temperatures and increasing fisheries pressure (*Gibson et al., 2023*). Meanwhile, there is growing concern that maintaining sustainable menhaden harvest limits and whole-ecosystem sustainability may be impeded by climate-related changes (*Anstead et al., 2021*; *Drew et al., 2021*). If post-breeding royal terns from the mid-Atlantic Coast region depend on climatically driven fish resources, especially during the costly life stage of migration, then any sudden shifts in the distribution and availability of these food resources could be catastrophic, as it was for royal terns in California feeding on pacific anchovy (*Engraulis mordax*) (*Buckley, Buckley & Mlodinow, 2021*).

Interest and investment in offshore renewable energy sources has increased globally, although at a lesser rate along the Atlantic Coast of the United States (*Musial et al., 2023*; *Sadorsky, 2021*). There remains a lack of understanding how these offshore facilities, many with an operational life of several decades, may affect seabird populations in the long term, as their presence and operation may potentially influence the selection of migratory routes, introduce collision risk, and affect access to important resources (*Loring et al., 2017*; *Searle et al., 2023*). Recent evidence suggests that these facilities will attract fish (*Knorrn et al., 2024*; *Wilber et al., 2022*), which in turn could attract larger terns to the areas (*Thaxter et al., 2024*). This knowledge gap makes understanding species space use in an area of proposed offshore energy development paramount, and these data are more accessible given advances in GPS-tracking technology and other survey methods (*Searle et al., 2023*).

The goal of this study was to describe the post-breeding season movements of adult royal terns from the largest seabird colony in Virginia, U.S.A. Using tracking data, we estimated the time the birds spent in 'resting' (typically non-flight activity) *vs.* 'exploratory' (*i.e.,* flying, foraging) states to understand behavior during this period. These are the first tracks of their kind for this species, and understanding these movements may determine what factors drive future population change (*Brinker et al., 2007*; *Gibson et al., 2023*).

## METHODS

### Study area

We studied the post-incubation movements of adult royal terns between 16 June–01 December 2022 that were captured and deployed with tracking devices on Rip-Raps Island, the site of Fort Wool in coastal Virginia (Fig. 1). Rip-Raps Island was altered to create suitable nesting habitat for a seabird breeding colony displaced by a massive transportation project on the South Island of the Hampton Roads Bridge-Tunnel (*Catlin et al., 2024*). Rip-Raps Island is attached to the South Island by a rock jetty. The former South Island colony was the largest royal tern colony in Virginia, supporting several other seabirds including common terns (*Sterna hirundo*), sandwich terns (*Thalasseus sandvicensis*), gull-billed terns (*Gelochelidon nilotica*), black skimmers (*Rynchops niger*), laughing gulls (*Leucophaeus atricilla*), American herring gulls (*Larus smithsonianus*), and great black-backed gulls (*Larus marinus*) (*Watts & Paxton, 2014*; *Watts et al., 2019*).

### Field methods

We captured adult royal terns between 16 June 2022–1 July 2022 using slip knot traps (knots made of monofilament line attached to an approx. 0.1 m$^2$ piece of wire mesh) placed near nests to capture individuals as they walked onto or off the nest after being displaced during trap setup. We deployed the tags near the end of the colony's incubation period after most nests had hatched. Five of the nine adults that were outfitted with tracking devices had been previously banded as chicks on South Island in 2018 ($n = 3$) and 2019 ($n = 2$); the ages of the other tagged birds were unknown as they were unmarked at capture. Royal terns generally do not return to breed until at least age 3, so this was likely the first or second breeding season for those five previously marked individuals (*Buckley, Buckley & Mlodinow, 2021*; *Gibson et al., 2023*). We were unable to determine the success of the nesting attempts of the birds that we tagged. We captured these birds just prior to the colony entering a communal crèche and subsequently fledging relatively synchronously. The bulk of the colony was fledged by end of June or beginning of July each year, marking the end of the breeding season and the beginning of migration. Through other work, approximately 3% of the colony was banded.

Once captured, royal terns were banded with a United States Geological Survey incoloy metal band and a white field-readable Darvic plastic leg band with a black engraved unique three-digit code (Interrex, Lodz, Poland). Each bird was weighed and a solar-powered GPS/GSM transmitter (ES-420 model; Cellular Tracking Technologies, New Jersey, USA, 10.3 ± 0.1 g (1 SD) total assembly weight, 2.4 ± 0.1% (range: 2.2–2.5%) body weight, 428 ± 20 g, range: 400–467 g) was affixed to the body with a legloop harness mount

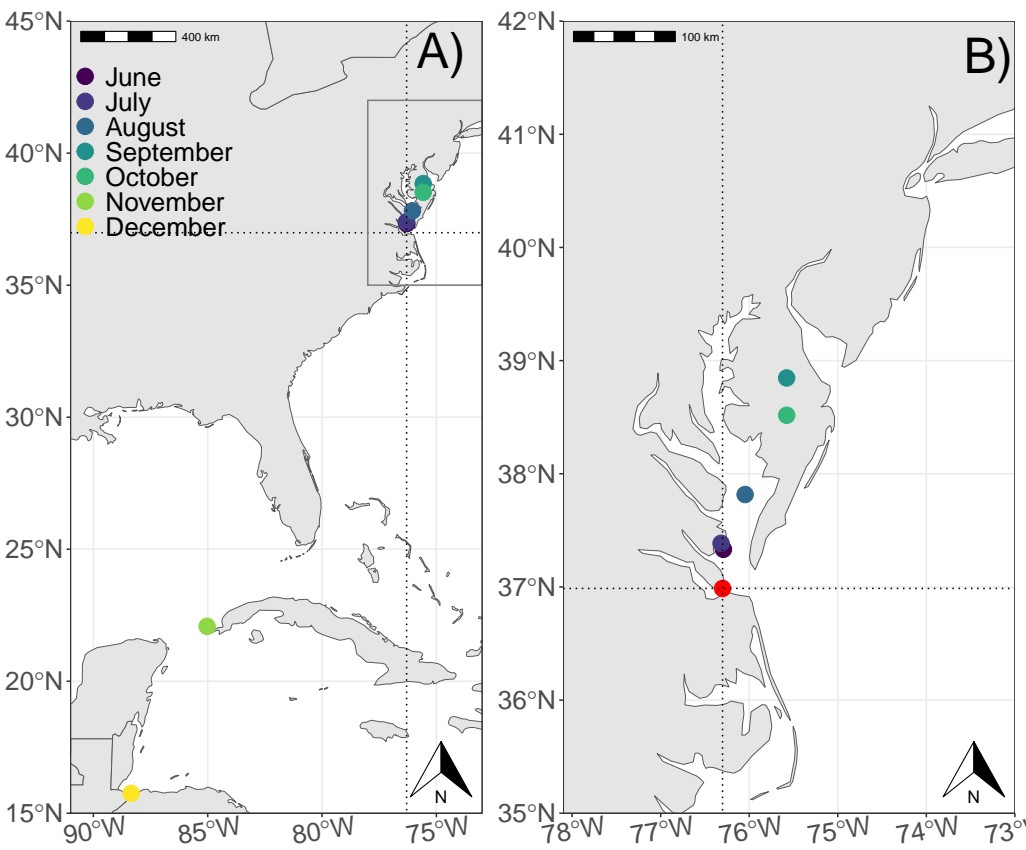

**Figure 1** **Map of the study area and average locations of royal terns by month.** (A) Average locations of royal terns from 16 June–01 Dec 2022 and (B) the Chesapeake Bay (inset from panel A) near the tagging location (36.9866, −76.30118). The points represent the monthly average latitude and longitude for tagged birds. We marked the site of the colony in panel B (red dot). The two dotted lines also cross at the colony site in both panels. Made with Natural Earth. Free vector and raster map data @ naturalearthdata.com.

made of $\frac{1}{4}$'' teflon ribbon (Bally Ribbon Mills, Pennsylvania, USA) and two copper crimp connectors that were used to fix the position of the ribbon and keep it from slipping. The transmitters were programmed remotely to record positions and instantaneous speed (km/h) every 60–120 min, seven days per week, battery life permitting. The variable fix rate was used to maximize battery life. A fix is the estimated two-dimensional (latitude, longitude) location of the unit calculated using at least four satellites. Data were collected until the transmitters stopped working or were recovered. We searched for tagged birds 2–3 times a week to assess impacts of the transmitters. We also monitored the tags to ensure that they were moving, and if not, we would investigate the cause.

## Modeling overview

We modeled the patterns of movements of individual royal terns in a hidden Markov model (HMM) framework using the momentuHMM package (*McClintock & Michelot, 2018*) in R (version 4.4.1, "Race for your life"). HMM-based approaches analyze individual

movement data to decompose time-series location data (*e.g.*, telemetry information) into direction, speed of travel, and turning angle between successive positions, using variation in these movements, and other covariates provided, to assign individual positions to an unobservable behavioral state (*Catlin et al., 2024*).

As our interest was in general patterns of space use, we developed a simple model to distinguish between two states: exploratory and resting. We acknowledge that the underlying behaviors of these states may change throughout the season, such that our results should be interpreted as general 'active' and 'inactive' states. Individuals were allowed to move freely between the two states. From band resightings and recoveries, we know that, following the breeding season, royal terns disperse throughout the region, foraging and resting in a variety of locations (*Gibson et al., 2023*). We assumed that birds in the exploratory state would have longer and more variable step lengths (distance between successive locations) and have more concentrated turning angles (tendency to maintain course between successive location) (*Catlin et al., 2024*).

## Data preparation

Although each transmitter was set to collect data at regular intervals, some of the possible observations were missing due to GPS fix issues and duty-cycle assignment. We created a dataset for hourly locations for each bird from the time the transmitter was affixed until the transmitter stopped working, or was otherwise known to be detached from the bird. For missing observations, we imputed the location information (*McClintock & Michelot, 2018*) using a continuous-time correlated random walk model using the 'crawlWrap' (*Johnson et al., 2008*) and assuming a bivariate normal measurement error model. We created 100 separate realizations that were used to create pooled estimates that accounted for location uncertainty, using the 'MIfitHMM' function. Locations within 500 m of the breeding colony where step length from the previous location also was <500 m were *a priori* assigned to the resting state, under the assumption that their behavior under these conditions would reflect resting behavior, even resting outside of the colony area. These known states were supplied to the model to aid in model-fitting and state assignment. We used two data streams for our modeling: step length and turning angle.

## Model

We assumed step length at each time step followed a gamma distribution with a mean and standard deviation (SD) that are estimated from the data (starting values: mean = 0.1 km, SD = 10 km). We assumed that individual turning angles (*i.e.,* short-term persistence in the path of travel) followed a wrapped Cauchy distribution with a mean of 0 (*i.e.,* no change from prior path of travel) and a concentration that is estimated from the data (starting value = 0.01), where higher concentration values indicate lower variance around the mean (circular–linear regression model, *McClintock & Michelot, 2018*). Pooled estimates were created with the 'MIpool' function, and we used the Viterbi algorithm to assign state probabilities, which is default for the momentumHMM package (*McClintock & Michelot, 2018*). Data underlying this manuscript are made accessible through the Virginia Tech Data Repository at https://doi.org/10.7294/27182037.

### Ethics statement

This research was completed under authorization of the US Geological Survey Federal Master Bander permit #21446, VDWR Scientific Collection and Bird Banding permit #2558265, and Virginia Tech IACUC protocol #19-248. The island where the captures were made is owned by the Commonwealth of Virginia and managed by the VA Department of Transportation. All research complied with all applicable laws and regultations.

## RESULTS

We captured and tagged 11 adult royal terns from 17 June–02 July, 2022. For one bird, the tag was removed the day after tagging as despite no obvious malfunction in the tagging process, the bird was not flying when we observed it the next day but did begin flying normally after the tag was removed. No other birds appeared affected by the transmitters. A second tagged bird was found depredated less than one month after tagging, after remaining in one location for five days. We could not determine the cause of the predation from remaining field evidence. We recovered the leg bands, small parts of the carcass, and the GPS unit under an avian predator perch. Thus, we analyze data from nine of 11 tagged birds.

From these nine birds, we collected 19,316 hourly locations. Individuals carried transmitters for 30–167 days, during which they traveled from 4,695–35,912 km (mean: 152–243 km d$^{-1}$; Table 1, Fig. 2). Individuals made 61–323 exploratory trips (defined as uninterrupted locations where the bird was designated as 'exploratory'), averaging 1.31–2.55 trips d$^{-1}$ (Table 1). Birds were tracked as far north as Jamaica Bay in Long Island, NY (40.63885°, −73.84400°). Although we truncated our analysis to December 1, 2022, one bird was tracked until February 2023, traveling 51,072 km in 242 days. Its last known location was near Puerto Cortes, Honduras (15.81407°, −87.94353°; Fig. 3). We observed five royal terns return to the colony, and an additional two were observed elsewhere the following year. The average instaneous speed recorded was 3.1 ± 9.8 km/h for resting birds and 19.5 ± 19.5 km/h for exploratory birds. The model fit the data reasonably well (Fig. S1).

Royal terns spent a greater proportion of their time in the exploratory state (mean: 0.711, 95% CI [0.710–0.712]) than in the resting state (mean: 0.289, 95% CI [0.280–0.290]), and the proportion of locations designated exploratory increased through the study (0.49–1.00; Table 2), but the number of birds sampled was variable, and only a single bird comprised the tracks in the final month of the study (Table 2). The average length of exploratory trips, increased through the study, from 54 km/trip during the breeding season to 240 km/trip during migration (Table 2). Mean exploratory trip length, duration, and the distance from the colony increased through the study (Table 2), and mean latitude and longitude indicated that birds moved to the northeast until September or October, and moved to the southwest after (Fig. 1).

Average step length for individuals in the exploratory state (10.7 ± 8.1 km) was greater than for those at rest (0.08 ± 0.05 km; Fig. S2). Individuals in the exploratory state showed higher short-term persistence in the path of travel (concentration = 0.19 ± 0.01) than those in the resting state (<0.001, Fig. S3).

Table 1   **Individual royal tern tracking statistics.** Royal terns were tagged at the end of the incubation period, 16 June–1 July 2022, on Rip-Raps Island, Virginia and then tracked with GPS/GSM transmitters from June 16–December 1, 2022. We used a hidden Markov model to distinguish between two behavioral states, exploratory and resting. A trip was defined as uninterrupted locations where the bird was designated as 'exploratory'.

| Individual | Deployed | Last signal | Distance traveled (km) | Days Tracked | Km d$^{-1}$ | Exploratory trips | Trips d$^{-1}$ |
|---|---|---|---|---|---|---|---|
| 1[*] | 17-Jun 2022 | 01-Dec 2022 | 35,912 | 167 | 216 | 297 | 1.78 |
| 2 | 17-Jun 2022 | 26-Oct 2022 | 24,350 | 131 | 186 | 172 | 1.31 |
| 3 | 17-Jun 2022 | 01-Nov 2022 | 21,705 | 137 | 158 | 218 | 1.59 |
| 4 | 17-Jun 2022 | 26-Aug 2022 | 11,010 | 70 | 157 | 163 | 2.33 |
| 5 | 02-Jul 2022 | 08-Nov 2022 | 24,070 | 129 | 187 | 323 | 2.50 |
| 6 | 02-Jul 2022 | 13-Aug 2022 | 8,256 | 42 | 197 | 92 | 2.19 |
| 7 | 02-Jul 2022 | 09-Sep 2022 | 10,903 | 69 | 158 | 167 | 2.42 |
| 8 | 02-Jul 2022 | 02-Aug 2022 | 4,695 | 31 | 152 | 79 | 2.55 |
| 9 | 02-Jul 2022 | 01-Aug 2022 | 7,186 | 30 | 243 | 61 | 2.03 |

**Notes.**
  *This individual's transmitter collected locations until February 2023, travelling 51,072 miles over 242 days.

Table 2   **Monthly tracking statistics of exploratory behavior.** Data are from nine royal terns that were tagged at the end of the incubation period, 16 June–1 July 2022, on Rip-Raps Island, Virginia; statistics derived from a hidden Markov model that distinguished between two states, exploratory and resting. A trip was defined as uninterrupted locations where the bird was designated as 'exploratory'. Data were summarized by trip and bird ID, then again by bird.

| Month | Individuals | Total hours tracked | Exploratory hours (proportion) | Mean trip length (±1 SD) | Mean trip duration (±1 SD) | Distance to colony (±1 SD) |
|---|---|---|---|---|---|---|
| June | 4 | 1,297 | 637 (0.49) | 54 ± 48 km | 4.6 ± 6.4 h | 54 ± 40 km |
| July | 9 | 6,518 | 3958 (0.60) | 81 ± 114 km | 6.6 ± 11.9 h | 54 ± 49 km |
| August | 9 | 4,663 | 3439 (0.73) | 81 ± 101 km | 8.0 ± 11.9 h | 85 ± 98 km |
| September | 5 | 3,078 | 2481 (0.81) | 107 ± 198 km | 12.2 ± 23.4 h | 194 ± 101 km |
| October | 4 | 2,843 | 2413 (0.85) | 150 ± 499 km | 15.3 ± 60.2 h | 155 ± 48 km |
| November | 3 | 912 | 803 (0.88) | 240 ± 460 km | 18.7 ± 28.0 h | 1,215 ± 1188 km |
| December | 1 | 5 | 5 (1.00) | 45.3 km | 5 h | 2,661 km |

## DISCUSSION

Royal terns traveled long distances, became increasingly exploratory through the study (Table 2), and they moved north prior to migrating south. Banded individuals from this colony have been resighted or recovered from New England, U.S.A. to Peru during the winter (*Gibson et al., 2023*), and one of the birds we tracked spent the non-breeding season in Honduras, illustrating the lengthy journeys these birds undertake. Migration is one of the costliest periods in a bird's life; as such, migratory behavior should reduce the time spent in migration (*Klaassen et al., 2012*). Thus, the northward post-breeding movements of royal terns that nested in Virginia must confer some benefit to individuals who employ this strategy. Here, royal terns did not begin to move south of the breeding colony until October, whereas birds banded in a Georgia, U.S.A. colony, did not move north prior to migrating south (D Catlin, 2019, unpublished data).

Prior work has demonstrated that royal terns forage primarily in shallow, inland waters up to 80 km from their breeding colony (*Buckley & Buckley, 1972*), and during the late

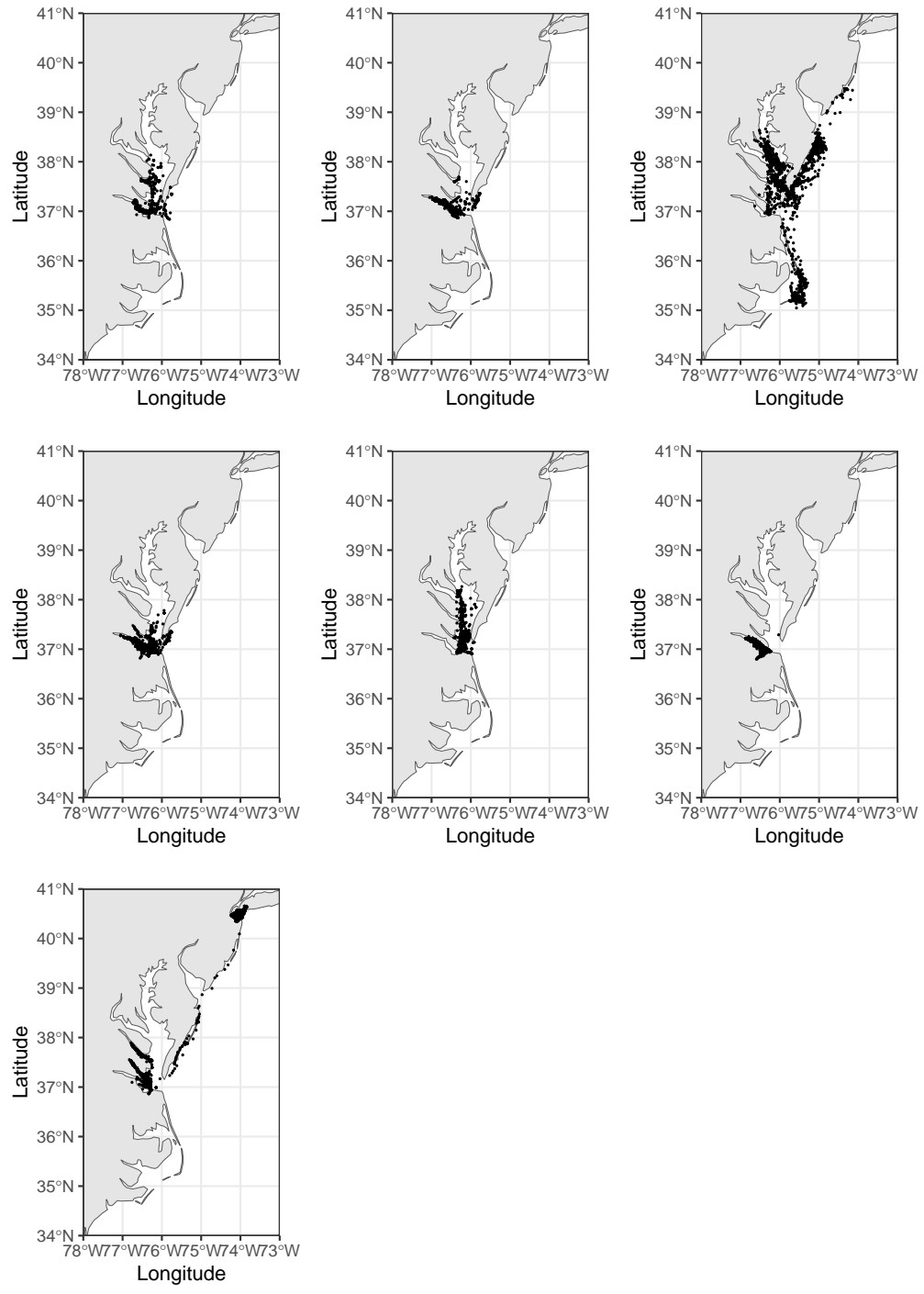

**Figure 2 Tracks of seven royal terns captured on Rip-Raps Island, VA, USA.** These birds remained above the 34th parallel while being tracked. Locations are from a GPS/GSM transmitter or imputed using a biased random walk estimator. Individuals were given transmitters in June–July 2022, and we tracked their movements until December 1, 2022. Made with Natural Earth. Free vector and raster map data @ naturalearthdata.com.

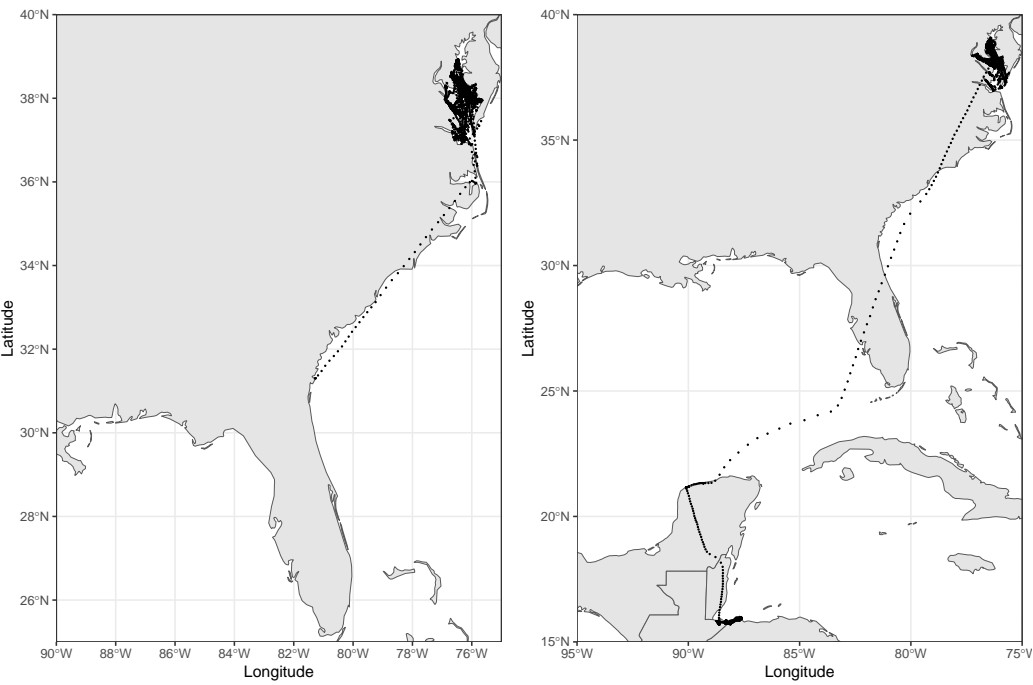

**Figure 3** **Tracks of the remaining two royal terns.** These birds were captured at the end of the incubation period, 16 June–1 July 2022, on Rip-Raps Island, VA, USA. These birds traveled below the 34th parallel while being tracked; locations truncated to Dec 1 2022. Locations are from a GPS/GSM transmitter or imputed using a biased random walk estimator.

incubation stage, individuals foraged <15 km offshore (*Gatto et al., 2019*; *Rolland et al., 2020*). During the late incubation and chick-rearing period (June), we found that on average individuals stayed within 54 km of the colony (Table 2). The average distances from the colony that we estimated were greater than the previously mentioned studies, which may indicate that royal terns nesting on Rip-Raps Island had to travel farther due to prey availability or quality (*Rolland et al., 2020*). Alternatively, the colony was on an engineered island, such that the birds were constrained to travel farther than they would from a natural site. Technology allowed us to track royal terns farther and longer than previous studies, so it is possible that distances increased as chicks grew larger and could survive longer periods without supervision (*Buckley, Buckley & Mlodinow, 2021*). *Rolland et al. (2020)* attempted to recapture and retrieve GPS trackers after three days and *Gatto et al. (2019)* used very high frequency (VHF) radio telemetry to track individuals for 11 days. Comparatively, we were able to track individuals for 30–169 days (Table 1). We also captured relatively late-nesting individuals, which is indicative of renests or young birds that may have lower hatching success (*Buckley, Buckley & Mlodinow, 2021*). Indeed, five of the nine individuals that we fitted with transmitters were first- or second-time breeders. Therefore, the individuals we captured late in incubation and subsequently tracked may never have hatched chicks, releasing them from the need to return to the colony as frequently.

*Buckley & Buckley (1972)* recorded royal terns traveling up to 65–80 km h$^{-1}$ parallel to their vehicle on a bridge over the Chesapeake Bay. Although our average recorded royal tern flight speeds were far slower, the transmitters did record birds traveling up to 100 km h$^{-1}$, albeit rarely. The average speeds that we recorded were less than half the speed of lesser black-backed gulls (*Larus fuscus*) during migration (38.6 km h$^{-1}$) (*Klaassen et al., 2012*) and for royal terns breeding in Louisiana (35.6 km h$^{-1}$) (*Rolland et al., 2020*). Both of those studies removed speeds of less than 10 km h$^{-1}$, but they used that speed to distinguish between resting and active states, rather than assigning states probabilistically, as in this study. Despite the relatively high instantaneous speeds, lesser black-backed gulls had relatively slow overall migration speeds that were attributed to frequent stops. Thus, lesser black-backed gulls, a relatively short-distance migrant, may have been employing an energy reduction strategy rather than a time-minimizing tactic (*Klaassen et al., 2012*). Royal terns in this study did not travel at faster instantaneous speeds during migration (Table 2), but, time in exploration and trip length increased.

Adult royal terns care for and feed their young throughout the day and night (*Buckley, Buckley & Mlodinow, 2021*), returning to feed every 2.5 h on average (*Erwin, 1977*), and royal tern chicks continue to beg and be fed through migration (*Buckley, Buckley & Mlodinow, 2021*). These extended efforts are reflected in other breeding strategies such as delayed age at first breeding and a single-egg clutch (*Buckley, Buckley & Mlodinow, 2021*). Moreover, fledged juvenile royal terns weigh less than adults captured during incubation (CE Weithman, 2022, pers. obs.), indicating that significant growth still is needed to achieve enough mass to survive migration. Given these somewhat strict constraints, royal terns ought to be capitalizing on a food resource (*Thorup et al., 2010*) that enables the birds to prepare for flights to wintering locations to the south (*Warnock, 2010*). *Quillfeldt et al. (2010)* found that thin-billed prions (*Pachyptila belcheri*) migrated south during the austral winter and that this behavior has increased in prevalence since the early twentieth century, presumably because of warming sea temperatures and in search of food. Such migrations (*i.e.*, into colder latitudes/altitudes) are exceedingly rare (*Newton, 2010*), but they show the flexibility of migration strategies, and the capacity for a species to adjust quickly to changing conditions (*Quillfeldt et al., 2010*; *Thorup et al., 2017*). We speculate that the northward movement of royal terns that nest in the coastal mid-Atlantic region of the United States is a local adaptation to their food resource. Migration itself is a form of resource tracking that can span hemispheres and continents.

We suggest that adult royal terns were preparing for a potentially lengthy migration south by tracking the seasonal northward flux of fishes (*Liljestrand, Wilberg & Schueller, 2019*; *Lucca & Warren, 2019*; *Nicholson, 1971*; *Nicholson, 1978*). Atlantic menhaden are an important food source for royal terns in the mid-Atlantic region (*Buckley, Buckley & Mlodinow, 2021*). *Gibson et al. (2023)* found that the indirect impacts of menhaden fisheries drove adult survival lower as fish production decreased. As such, the spawning and distribution of menhaden may affect royal tern post-breeding distribution and movements. During the royal tern breeding season, Atlantic menhaden are distributed from Maine to Florida, but the largest menhaden are found farther north in the Atlantic and Chesapeake Bay (*Ahrenholz, 1991*; *Liljestrand, Wilberg & Schueller, 2019*; *Lucca &*

*Warren, 2019*; *Nicholson, 1971*; *Nicholson, 1978*). This distribution remains stable until September when the fish begin to migrate south (*Liljestrand, Wilberg & Schueller, 2019*; *Lucca & Warren, 2019*; *Nicholson, 1971*; *Nicholson, 1978*). By December most schools are south of the Chesapeake Bay, and schools disperse by January (*Ahrenholz, 1991*; *Nicholson, 1978*). Thus, we suspect that royal terns in this study moved north to find larger or more available prey and then migrated with these schools as they moved south. Resource tracking by migratory animals is relatively common (*Thorup et al., 2017*). This pattern of large-scale northward migration was absent from a banding study of royal terns in coastal Georgia, but their movements and locations (D Catlin, 2019, unpublished data) were superficially similar to those of the Gulf menhaden (*Brevoortia patronus*;), who tend to migrate not latitudinally, but between off-shore and near-shore waters (*Ahrenholz, 1991*; *Vaughan, Shertzer & Smith, 2007*).

## CONCLUSION

Little is known about the post-breeding movements of royal terns but understanding their resource needs and behavior at this critical period could vastly improve the success of any restoration efforts. Like many terns, the location and abundance of food resources can have an important effect on population persistence (*Gibson et al., 2023*). As pacific anchovy (*Engraulis mordax*) stocks declined, the Pacific breeding royal terns all but disappeared (*Buckley, Buckley & Mlodinow, 2021*), suggesting that the anchovies were critical to the presence of royal terns on the North American Pacific coast. Similarly, *Gibson et al. (2023)* found that survival of adult and pre-breeding (ages 0–2 years) were affected by climate and indirectly through decreasing Atlantic menhaden production. The dependence of royal terns on menhaden and the relationship between menhaden and sea temperatures may partially explain the negative correlation with sea surface temperatures and royal tern survival (*Gibson et al., 2023*).

Additionally, several offshore wind energy lease areas along the mid-Atlantic Bight (*BOEM, 2025*), where we recorded tracks of tagged royal terns, are moderate to high-importance habitat to Atlantic menhaden. It is unclear if construction activity in the lease areas will dispel (*e.g.*, noise and disturbance during construction) and/or attract fish populations (*e.g.*, accumulation of invertebrates on structures that create artificial reefs) (*Friedland et al., 2021*), but it seems likely that they will attract fish (*Knorrn et al., 2024*; *Wilber et al., 2022*). For example, GPS-tagged sandwich terns showed more complex interactions with bird and prey species' space use in and around offshore wind farms than simple avoidance or range shifts (*Thaxter et al., 2024*). With the concern over Atlantic menhaden stock viability (*Anstead et al., 2021*; *Drew et al., 2021*) and other ecosystem interactions on the horizon, studies like these will be necessary to monitor and maintain viable royal tern populations in the face of uncertainty.

## ACKNOWLEDGEMENTS

We thank Donald Lyons and Keenan Yakola for their guidance and advice on GPS unit harnessing techniques. Thank you to Daniel Gibson for assistance during data collection

and providing feedback on manuscript development. We thank A. Caravaggi, E. Nol, E. Jones, and an anonymous reviewer for their review of the manuscript. Thank you to Donald Fraser for logistical support. We are thankful to David Norris and additional staff for their communication and logistical support during data collection. Virginia Tech acknowledges that we live and work on the Tutelo/Monacan People's homeland, and we recognize their continued relationships with their lands and waterways. Virginia Tech also acknowledges that its Blacksburg campus sits partly on land that was previously owned by families that also owned enslaved people. Enslaved Black people generated resources that financed Virginia Tech's predecessor institution, the Preston and Olin Institute, and they also worked on the construction of its building.

### Funding

Transmitters were funded through the U.S. Army Corps of Engineers, Environmental Management Restoration and Research program (Project No. 485301) (Task No. A1000). Fieldwork was funded by the Virginia Department of Wildlife Resources. The funders had no role in study design, data collection and analysis, decision to publish, or preparation of the manuscript.

### Grant Disclosures

The following grant information was disclosed by the authors:
U.S. Army Corps of Engineers, Environmental Management Restoration and Research program: 485301.
Virginia Department of Wildlife Resources.

### Competing Interests

The authors declare there are no competing interests. Michael Guilfoyle is an employee of the U.S. Army Corps of Engineers.

### Author Contributions

- Chelsea Weithman conceived and designed the experiments, performed the experiments, analyzed the data, prepared figures and/or tables, authored or reviewed drafts of the article, and approved the final draft.
- Daniel Catlin conceived and designed the experiments, analyzed the data, prepared figures and/or tables, authored or reviewed drafts of the article, and approved the final draft.
- Sarah Karpanty conceived and designed the experiments, performed the experiments, authored or reviewed drafts of the article, and approved the final draft.
- Kelsi Hunt conceived and designed the experiments, performed the experiments, authored or reviewed drafts of the article, and approved the final draft.
- Camille Alvino performed the experiments, prepared figures and/or tables, authored or reviewed drafts of the article, and approved the final draft.

- Joanna Morelli performed the experiments, authored or reviewed drafts of the article, and approved the final draft.
- Will Britton performed the experiments, authored or reviewed drafts of the article, and approved the final draft.
- Ruth Boettcher conceived and designed the experiments, performed the experiments, authored or reviewed drafts of the article, and approved the final draft.
- Rebecca Gwynn conceived and designed the experiments, performed the experiments, authored or reviewed drafts of the article, and approved the final draft.
- Michael Guilfoyle conceived and designed the experiments, authored or reviewed drafts of the article, and approved the final draft.
- James Fraser conceived and designed the experiments, performed the experiments, authored or reviewed drafts of the article, and approved the final draft.

### Animal Ethics

The following information was supplied relating to ethical approvals (i.e., approving body and any reference numbers):

This research was completed under authorization by the Virginia Tech Institutional Animal Care and Use Committee (19-248).

### Field Study Permissions

The following information was supplied relating to field study approvals (i.e., approving body and any reference numbers):

Virginia Department of Historical Resources and Virginia Department of Transportation coordinated access to Fort Wool which was granted under VDWR Permit #2558265 in coordination with these additional state agencies. Field work was also permitted under U.S. Geological Survey Federal Master Bander permit #21446.

### Data Availability

Data is available at Figshare:

Catlin, Daniel; Karpanty, Sarah; Weithman, Chelsea (2024). Data for "Seasonal movements of post-breeding royal terns (*Thalasseus maximus*) in Virginia". University Libraries, Virginia Tech. Dataset. Available at https://doi.org/10.7294/27182037.v2.

### Supplemental Information

Supplemental information for this article can be found online at http://dx.doi.org/10.7717/peerj.19898#supplemental-information.

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
