# Peer review of "Movements of post-breeding royal terns (Thalasseus maximus) in Virginia, U.S.A"

_PeerJ, doi:10.7717/peerj.19898_

## Round 0.1 · original submission · Major Revisions

Dear author,

We were fortunate to recieve three detailed reviews that offer the opportunity for further development of your work. Reviewers two and three in particular have noted informational gaps and methodological and inferential issues that must be addressed.

Best,
Anthony

·

Basic reporting

No comment.

Experimental design

No comment.

Validity of the findings

No comment.

Additional comments

Review of PeerJ paper. ROTE migration.
Nice to see the fruits of difficult labour tracking seabirds. I make some comments about your primary conclusion that the birds may be tracking Atlantic Medhaven but otherwise I found the paper interesting and using the hidden Markov method for identifying behaviour. I would like clarification on how the periods were determined, and also perhaps less emphasis on the results of the method, and more on the significance of the tracks themselves, given the fairly small sample size (but with great coverage for some individuals).
1. Abstract: State where Virginia is at first mention, as the state name may not be as familiar to international readers.
2. L. 44. I would break sentence into two and omit the word ‘however’ in the middle.
3. L. 47. Add ‘in birds’ after ‘migration strategies’
4. L. 73. Explain in what geographic area this catastrophic decline occurred.
5. L. 75. I think that you do not need the word ‘Additionally’ here. It is a new paragraph so reader does expect a different topic.
6. L. 77. I personally do not like the overuse of the word ‘however’ and in many cases it is not needed as in this sentence.
7. L. 82. I know that there is some debate about whether ‘data’ is a plural or singular word…I am still in favour (or very emphatically trained!) to use it in the plural form but realize that this usage may be changing. (see L. 128 and elsewhere where this word is used in plural form).
8. L. 86. Add USA after Virginia.
9. L. 100. Delete ‘is’
10. L. 102. Island is ‘Rip rap’ elsewhere ‘Rip-raps’. On figure 1 it is ‘Rip rap’ but other figures ‘Rip-raps’
11. L. 106. Herring gulls in North America are now American herring gulls. The scientific name has also changed.
12. L. 117. I think that you need to add here that they were previously marked as part of a long-term study where a large proportion (percentage?) of birds were individually identified with bands.
13. L. 180. An unclosed bracket.
14. L. 183. ‘Depredated’ is a better word here. [Predated is often used in anthropology]. Any idea of the predator?
15. L. 206. Have you stated how you differentiated post-breeding period movements from those you state are migratory or did you just assume migration when the birds turned southwards?
16. L. 219. First line of discussion mentions tracking fish but where are the data for this conclusion? The authors cite many other papers so the authors should probably state that they are assuming that they are tracking fish, as has been found in other studies (where presumably fish were also measured). I note that they use the word ‘appeared to’ but the reader will not know on what basis this conclusion is reached at this point (and without reference to these other papers). The first paragraph of the discussion usually reiterates the main findings but this paragraph does not do that. For example, I do not think that we know from the results that birds start to move south in October. I see that the Figure 2 caption defines these periods. I think it is worthwhile putting these dates into methods. There is much more in the tracks that can be mentioned in this first paragraph (e.g., use of Chesapeake Bay, see below).
17. L. 246. Why do the authors not know whether the outfitted birds raised chicks? This adds quite a bit of uncertainty to the periods defined as I suspect that birds would move much shorter distances if they were provisioning chicks. This limitation in interpretation should be made explicit in methods or results. See comment 15 above.
18. L. 262. Clarify which periods. Do the authors mean incubation, post-breeding and migration?
19. L. 277. Here the authors state that the idea of the birds tracking a food resource is speculation so I would advise changing the first paragraph of this discussion to acknowledge this (or just wait to discuss the idea of tracking food until later in discussion).
20. If there are such good data on menhaden migration is it possible for the authors to use it in their discussion so that the timing of movements can be more directly compared?
21. Figure 2. Is 1 July the end of all the terns’ breeding seasons or just the ones that had GPS tags affixed?
22. From Figure 4 it looks like only one individual was tracked south to its non-breeding grounds and a total of 4 were tracked turning south? I only see tracks of three going to lower latitudes in this figure.
23. Table 2. Are these the means of the individuals then averaged or the average of all tracks?
24. Somewhere in paper the authors need to acknowledge how few individuals were tracked north along the Atlantic but also that most birds that moved north went into the Chesapeake Bay. Is this where the menhaven go? Where there is a description of seasonal movements of the fish there is no mention of this large important bay. Clearly if the birds are tracking prey then the prey must be in the Chesapeake. More nuance to the results of the movements is needed. While the authors rely heavily on using the designation of movements as resting or exploratory it seems that just looking at the tracks and how long and where the birds are may be more promising in terms of its heuristic value in understanding post-breeding movements. From the description in the text of the movement of the fish it appears that only one bird might be tracking the menhaven up the Atlantic coast, with the majority using the Chesapeake, presumably also feeding there (unless the menhaven do go there too).

Reviewer 2 ·

Basic reporting

The manuscript is largely well written however, the language could be clearer – I have provided examples of where below.

At several places in the introduction the authors rely on references rather than including the point they want to make, and the flow of text could be improved, for example:

Line 39. This sentence is quite vague, why is this period critical or migration and what type of strategies are used? More detail would be useful. Also at the start of this paragraph you talk about all avian species but then only focus on seabirds at the end - it feels like a link is missing here to why you suddenly introduce seabirds? As studying this period of the annual cycle is particularly difficult in seabirds as they are typically away from land?

Line 52. Why particularly young birds? And it would be good to specify here that you mean juveniles and / or immatures rather young which is a more arbitrary term.

Line 68. It would be useful here to include the actual timeframe rather than 'decades' and also on line 62, to better understand when that negative relationship occurred and when that related to the Menhaden stock recoveries. Or do the authors means that the menhaden stocks have been up and down over this period? This is currently unclear.

Line 73. Again a little more detail would be useful here for readers not aware of this relationship. i.e. where and when did this occur and what was the impact on the terns.

Tables and Figures:

Please increase the size of labels and axis text in all figures, including those in the supp material. Also there are no figure legends that I could see for Supplementary figures 1 and 2.

Figure 2 - Please include the location of the breeding colony is. It would also be useful to show which fixes were assigned as exploratory and which as resting. And what does resting refer to - being at the colony as well as also resting during foraging trips / roosting whilst on migration? It is also good practice to include the HMM figure to show the classification output figure, to show the likelihood of an individual being in each state at a given point, which can be included as supplementary material.

Figure 3. Please include how many terns are contributing to the data in each month. More should be made of this plot in the results to demonstrate the slight movement north post-breeding - did all terns tracked head north before heading south? And how many individuals had data for long enough to show this behaviour.

Figure 4. Suggest removing this figure or adding to supplementary material. Also it looks from this figure that only one individual headed any distance north before heading back south?

Table 1. This definition of a trip is not typical so needs more explaining in the main text. During the breeding season a trip is usually defined as from when a bird leaves the breeding colony until it returns. again? From this table only 4 birds were included in the migration season so this should be emphasised in the main text.

Table 3. More detail is required to inform readers why these transition probabilities are important here and what they actually mean ecologically for the terns as this is not currently clear.

Table 4. The relevance of this table is also not clear? I would argue these are not results but just show how the HMM is modelling your data / assigning states?

Experimental design

The hypothesis are currently weak and are not well defined. More detail is required specifically on how the terns might ‘minimise their migration effort in other ways’ given that this is an important aim of your study. The authors also do not state clearly how this research fills an identified knowledge gap, especially as others have shown that the terns head north after the breeding season before migrating south.

I am also concerned that the authors so not include any acknowledgment of potential tag effects. Did the authors have any control birds (i.e. ringed but untagged terns) to look at tag effects i.e. return rates / breeding success? It is important that tag effects are considered as part of any tracking project to for welfare issues but also to ensure the resulting data is representative (i.e. the birds are undertaking normal behaviour). This is especially important given the behaviour you observed on the one individual whose tag was removed the day after tagging.

In places more detail is required in the methods, specifically:
Line 123. Was 2.4% the mean body weight of all individuals - please state this and also include the standard deviation.
Line 125. Need to clarify here what the copper crimps were connected to? They connected the harness to the device?
Line 126. Why was a range of 60-120 minutes used for the fix rate?
Lines 64 - 172. Please include here the starting parameter values for step length and turning angles you used in the model to define your two states? Also what method did you use to assign the most likely states, I assume you used the Viterbi algorithm so please state that here. Also please include which r packages and functions you used and cite them.

Validity of the findings

This study is largely descriptive with no statistical analysis therefore the findings of the study are difficult to interpret. The results as currently presented do not convincingly provide evidence for resource tracking as the authors state in the title, abstract and discussion.

The results and discussion also do not link back to the hypothesize made by the authors. I struggled unfortunately to see the relevance of much of the discussion. I think if the results section was stronger with statistical analysis to test for difference between seasons and potentially link the movements of the terns to the distributions of Atlantic Menhaden then the discussion should be easier to frame. At present the results do not provide any evidence for or even really look to answer the hypothesis put forward. From the title, one might expect some analysis looking at resource selection functions or linking the terns distributions to that of their prey.

Alternatively, I suggest reframing the manuscript entirely to focus on the more basic movements (i.e. trip / movement characteristics) of the terns during the difference seasons – as this is a fantastic dataset given that the terns have not been tracked for this length of time before. At present it is very unclear what the ecological significance is of stating that exploratory behaviour increases though post-breeding and migration as this would be expected once the birds are released from central place foraging. The authors also need to more clearly explain what the two HMM states actual mean in terms of the terns behaviour during each season i.e. does resting include brooding on the nest and roosting during migration?

Additional comments

INTRODUCTION
Line 53. To uses of the word 'favourable', please rephrase or use an alternative word for one i.e. 'access to more favourable wind current or migratory routes'. Also what avian predators might Royal Terns be at risk from?
Line 57. Suggest replacing 'but' with 'although'
Line 61. Suggest replacing 'imperil' with 'threatened' or similar
Line 65. It is noy clear here what the authors mean by 'Among other associations', it would be useful to rephrase so the meaning is clearer.
Lines 75-81. This paragraph comes a little out the blue after specifically talking about fish stocks. Are there references you can include here that suggests that Royal Terns might specifically be at risk from offshore renewable developments and in what way i.e. displacement of foraging habitat / barrier effects during migrations / collision risk? And is it likely that such developments will be located along the Atlantic Coast in the future?
Line 95. Would 'drive' be a more appropriate word here than 'may be linked' which is quite vague?

METHODS
Line 98. Missing 'between' before the dates
Line 99. 'Outfitted' doesn't seem quite right here - suggest changing/rephrasing and using 'deployed'. Also on line 113.
Line 100. Delete 'is' before 'was'
Line 110. Missing 'of' after 'made'?
Line 112. Remove 'very'
Line 115. Change 'was' to 'were'
Line 120. Include 'unique' before 3-digit - if that was the case?
Line 145. I don't think predicted is the right word here, assumed is probably more accurate?

RESULTS
Line 180. Delete ) after 2022.
Line 189. It is not clear what the authors mean by 'exploratory trips' here
Line 198 - 206. This would be expected?! As changed from central based forager to migrating. So these results are not saying anything new?
Line 207. This is not a result as this is what the HMM was set to do? Missing space between 0.08 KM. And the same for speed. Argue this is just part of explaining the HMM rather than results providing information on the terns behaviour.
The authors need to be clear throughout how many individuals are included in each season.

DISCUSSION
I have not reviewed this section extensively as I feel it needs to be reframed once the results section is strengthened. At present there is no clear story / flow to the text linking the intro, results or discussion.
Line 219. Your study did not really provide any evidence that the terns were preparing for a long migration of tracking fish? Some individuals did move north following the breeding season but the reason for that can only be assumed from the literature stated here not rom your results so this should be emphasised here.
Line 221. I am not clear how knowing birds spent most of their time in the exploratory state really tells us esp. as the authors don't interpret what this means i.e. during the breeding season this relates to foraging / chick provisioning? and post-breeding related to migration / dispersal away from the colony? If that is the case we would expect the birds to spend more time not resting as they no longer have a nest or chicks to protect?
Lines 231 - 247. Although this section provide interesting information on the Royal Terns is does not link back to the authors original hypothesise or title so feels out of place? This can also be said for lines 248 - 253?
Line 277. But from your introduction and start of the discussion this was already known for Royal Terns?
The discussion is currently lacking a critical review of the study’s limitations.

·

Basic reporting

Ethics approval
A federal bird banding permit and a research protocol review have been provided in the documentation with this manuscript submission

Data and R code
I downloaded the data and R code form the DOI supplied in the manuscript. The R code ran without issue. It required me to install two additional packages momentuHMM and doFuture

Figures
Fig. 3. – This figure is quite difficult to interpret. I can see that there is more data are more northern latitudes in months 8-10, and then a lot of data far south in month 11. However, it’s not clear how these data are distributed amongst individuals (for example, was it just one individual spending time at northern latitudes, or some, or all?). I think this figure needs to be rethought to portray relevant information and help the reader with interpretation.
Fig. 4 – This figure is nearly impossible to interpret and extract relevant information from. Why is there not at least colour-coding, labelling of seasons. Apart from seeing that one bird went far south in November, I don’t know how to interpret this further.
For Fig. S1, there are tracks shown that are moving over land in a straight line – I think these are interpolated tracks, and if so, should they not be masked for land?

Presentation
The manuscript is well written throughout and well referenced.

Experimental design

Tagging effects
It is noted in the paper that tag effects were observed for one bird (line 180). Was there any other either direct observation, or interrogation of the data to determine whether there may have been tag effects? From an ethics point of view, I think it’s important to state whether there were efforts to determine any detrimental effects to individuals.

Hypotheses
The central research objective is to ‘investigate the post-breeding season movements of adult royal terns’ and the authors gave two hypotheses in the manuscript:
1. (L91) ‘…adults would travel northwards away from the breeding colony spending time foraging and resting prior to their southward migration’
2. (L92) ‘…given this northward shift, royal terns would act to minimize their migration effort in other ways (i.e., speed, time spent migrating, etc.).
I think the first objective has been partially met. Certainly in Fig. 2, it is clear that the terns spent time north of their breeding colony. Perhaps I’m missing some information, but Figs. S1 and S2 do not have captions so I do not know what they are showing. However, it seems likely that Fig. S1 is showing all locations for each bird, and if this is the case, then there is heterogeneity amongst individuals that was not discussed in the paper, and which seems relevant considering the hypotheses being tested. Individuals are displaying different patterns of spatial behaviour, and although these tracks are not disaggregated by season, it would suggest that the strong northwards movement is primarily driven by two individuals (top right and bottom left).

The first hypothesis specifically states ‘foraging’ and much of the introduction and discussion focuses on this important behaviour. However, the HMM only has two states, which is ‘resting’ and ‘exploratory’. Exploratory includes foraging and travelling together in a single state, and the interpretation of ‘exploratory’ will change over time as birds transition between breeding (where they are likely provisioning chicks and so foraging for much of the time) through to post-breeding, and then on to migration where travelling will be more prevalent.

Including environmental information in the model, such as prey, if it is available, or bathymetry if it is not, may help to fit a 3-state model to distinguish foraging with travelling (e.g. using a 3-state model to investigate travelling, foraging, resting). This would be helpful in testing the first hypothesis, in whether these individuals moved northwards to forage (and rest).

It’s less clear from the results how the second hypothesis has been addressed in this study, except for text in the Discussion.

Modelling
• L 130 - The definition and specification of the HMM is not adequately described. Please refer to Langrock et al (2012) https://doi.org/10.1890/11-2241.1 or Zucchini, W., MacDonald, I. L., & Langrock, R. (2016). Hidden Markov models for time series: An Introduction using R (2nd ed.). Chapman and Hall/CRC for references on how to describe a HMM.
• I think speed of travel was used as well as turn angle and step length to investigate the second hypothesis. However, it is not explained in the formulation of the model, or in the results why this was included or the interpretation. Can you please provide an explanation in the text.
• Where any data discarded directly after tagging, to account for any temporary tagging effect (e.g. resting for a longer time than would be usual for an individual)? This may be important because these data were being used to fit the HMM so unexpected behaviour is likely to have an impact on model fit.
• Does ‘resting’ behaviour include resting at the colony, on land, and at-sea? If so, ‘resting’ may have a different definition in the breeding season (where they may almost exclusively be at the colony) compared to the migration season (where they are not at the colony, and resting at-sea/on land).
• Can you provide plots of step length and turn angle by state?
• Were any diagnostics on model fit, or model selection carried out? You could show residuals or a QQ plot here.

Validity of the findings

Discussion
The first sentence is (L219) ‘Adult royal terns appeared to be preparing for a potentially lengthy migration south by tracking the seasonal northward flux of fishes…’ but how do the authors know the birds are tracking fish, as this wasn’t presented in the analysis. If this is just inferred (from previous studies) then the wording needs to be rephrased to make this clear.
L 221 ‘Royal terns spent most of their time in the exploratory state…’ – whilst true, when disaggregated into season, in breeding season, they spent most of their time resting, and only in post-breeding and migration were they in an exploratory state for most of the time.
L222 ‘…and they became increasingly exploratory as migration approached and they moved south to their non-breeding grounds’ – where is this shown in the results?
L229 ‘Here, royal terns did not begin to move south of the breeding colony until October’. – I think this is trying to answer the second hypothesis, and if so, this statement needs to be evidenced about why this is important for this colony – has it been shown that surrounding colonies move south earlier?
L259 ‘Royal terns in this study did not travel at faster instantaneous speeds during migration (Table 2), but time in exploration and trip length increase. Thus, the amount of time spent in migration would have decreased through the periods, as would be expected for a long-distance migrant’. This is a key sentence for demonstrating that the second hypothesis is being addressed, but from the results presented, I would not have been able to come to this conclusion.

Additional comments

Minor comments
• L57 – ‘survival rates appear to be rebounding’ – Does this mean survival rates over all age classes, or is this unknown?
• L65-70 – To be clear, prey have undergone (cyclical?) declines but 0-2 age class royal terns have seen only sustained declines, or have they undergone periods of recovery as well?
• L82 – replace ‘this data’ with ‘these data’
• L100 – Rip-raps Island is was altered
• L123 – add in min and max % of body weight as well
• L136 – the text states the three defined periods are partially overlapping, but gives dates that are not overlapping, which is confusing
• L145 – ‘predicted’ I think should be replaced by ‘hypothesised’ or ‘expected’, as you’re not predicting from a model here
• L190 Jamaica Bay is not marked on Fig. 2
• L235 – “The average distances from the colony that we estimated were greater than the previously mentioned studies, which may indicate that royal terns nesting on Rip-raps Island had to travel farther due to prey availability or quality” Could this at least partly be because Rip-raps Island is an artificial nesting site and so has different characteristics to other (natural) colonies?
• L310 – Offshore wind developments have been mentioned several times in the paper. It would be useful if current/future developments were added on to a figure so that the reader can get a sense of how close these are to the colony/migratory routes of this species.
• Additionally, it would be useful to include maps of prey of this species in the paper, if these data are available.
• L319 – Either the word ‘uncertainty’ or the reference to Searle et al (2023) is used incorrectly here. ‘Uncertainty’ here is being used in the context that this species faces an uncertain future, whereas the citation is a paper on improving the treatment of uncertainty in assessments, which is a quantifiable measure. I suggest rephrasing.
• Fig. 1. – An inset map of the USA would be helpful for non-US readers. The maps appear blurred so I think they need to re-rendered for publication. A north arrow would be helpful, as would a scale bar.
• Fig. 2. – Can you add on the breeding colony and label clearly please. The figure is blurred so I think the images need to re-rendered for publication. It would be helpful if Fig. 2b and c could have insert boxes so that it’s clearer which part of (c) that (b) is showing, and which part of (b) that (a) is showing, if that makes sense?

---

## Round 0.2 · Minor Revisions

I agree with the reviewers that the edits have substantially improved the manuscript and that only minor issues remain. I do also have one specific request regarding the methodological details, echoing reviewer 2. Some information on how the end of the breeding season and start of migration were delineated would be useful, for clarity.

·

Basic reporting

No Comment.

Experimental design

No Comment.

Validity of the findings

No comment.

Additional comments

I have made just a few comments. These refer to minor formatting and editorial issues.
Note that at least four of the references still have capitalization throughout the titles (e.g Rolland et al, Nicholson et al, Lindstrom et al, Gatto et al.)
L. 25. It seems odd that the upper CI contains the mean. Is this just a function of rounding?
L. 25-26. Make more efficient by removing 'a food resource' and add 'the' in front of Atlantic Menhaden.
L. 313. Can you rephrase as you use the words 'We speculate' just above. How about 'We suggest...
L. 235. It appears that the authors have removed most references to the stage at which the birds might have been. Is the mention of the breeding period here an oversight or did the authors know that this bird still had eggs or chicks?
No copy of Figure 3 on PDF (although it was in materials sent).
There are references to Tables 3 and 4 but I do not see these on PDF.

Reviewer 2 ·

Basic reporting

All looks good, the manuscript is clear and well written. The updated figures also look great, and are much clearer to read.

Experimental design

Northing further to add.

Validity of the findings

Much clearer now the scope of the manuscript has been changed.

Additional comments

Thanks very much to the authors for addressing my previous comments so comprehensively, and those from the other reviewers. The change to focus on the tern’s movements more broadly, rather than the focus on resource tracking, has made the manuscript much stronger.

I just have a few small additional comments, with line numbers referring to the track changed version.

Abstract
Line 32. Now that you have a little more room in the abstract it would be useful to include here examples of these threats i.e. declining prey fish / offshore renewables as included in the conclusion.

Methods
Line 141. add 'the cause' after 'we would investigate'

Results
Line 205. This added sentence (The model fit the data reasonably well (Fig. S1). ) feels out of place here - was it meant to be added elsewhere?
Line 223. Retain 'of' here after 'length'
Line 224. These periods are no longer defined in the methods but it would be useful to really briefly include in the methods how you defined the end of the breeding season and start of migration (i.e. end of returning to a central place?). i.e. providing the range of dates (or at least the month) the tacked terns finishing breeding / started migration would be useful to then see how this aligns with the data in Table 2
Line 234 and 236. Delete 'and was similar among periods, except that step length was longer for exploratory birds during the breeding period than in the other two periods (Table 4)' now that these periods were not included in the analysis.

Discussion
Line 259. It is not clear here if the authors mean that the Georgia ringed birds didn't make a northly movement before heading south, or that they made a northerly movement but then didn't move south - so please clarify that here.
Line 284. Delete 'regardless of the period (Table 2)’ here as period was no longer included in your analysis.
Line 292-294. I am not sure I understand this sentence, especially with the added text. To me, this reads as the time spent in migration would decrease from incubation to migration, which I don't think the authors mean?

---

## Round 0.3 · accepted · Accept

Dear authors,

Thank you for submitting your revision. I am happy to say that I can now recommend the work for publication. Many congratulations.

Best,
Anthony